# Longitudinal Analysis of SARS-CoV-2-Specific Cellular and Humoral Immune Responses and Breakthrough Infection following BNT162b2/BNT162b2/BNT162b2 and ChAdOx1/ChAdOx1/BNT162b2 Vaccination: A Prospective Cohort in Naive Healthcare Workers

**DOI:** 10.3390/vaccines11101613

**Published:** 2023-10-19

**Authors:** Geon Young Ko, Jihyun Lee, Hyunjoo Bae, Ji Hyeong Ryu, Hye-Sun Park, Hyunhye Kang, Jin Jung, Ae-Ran Choi, Raeseok Lee, Dong-Gun Lee, Eun-Jee Oh

**Affiliations:** 1Department of Biomedicine & Health Sciences, Graduate School, The Catholic University of Korea, Seoul 06591, Republic of Korea; geonyouong0107@catholic.ac.kr (G.Y.K.); onion1002@catholic.ac.kr (J.L.); hjin@catholic.ac.kr (H.B.); 2Department of Laboratory Medicine, Seoul St. Mary’s Hospital, College of Medicine, The Catholic University of Korea, Seoul 06591, Republic of Korea; 22280094@cmcnu.or.kr (J.H.R.); 22180049@cmcnu.or.kr (H.-S.P.); 21803188@cmcnu.or.kr (H.K.); jiinj@catholic.ac.kr (J.J.); bibi@cmcnu.or.kr (A.-R.C.); 3Resesarch and Development Institute for In Vitro Diagnostic Medical Devices, College of Medicine, The Catholic University of Korea, Seoul 06591, Republic of Korea; 4Division of Infectious Diseases, Department of Internal Medicine, Seoul St. Mary’s Hospital, College of Medicine, The Catholic University of Korea, Seoul 06591, Republic of Korea; misozium03@catholic.ac.kr (R.L.); symonlee@catholic.ac.kr (D.-G.L.); 5Vaccine Bio Research Institute, College of Medicine, The Catholic University of Korea, Seoul 06591, Republic of Korea

**Keywords:** SARS-CoV-2, vaccine, humoral response, cellular response, breakthrough infection, ELISPOT, neutralizing activity

## Abstract

Assessing immune responses post-SARS-CoV-2 vaccination is crucial for optimizing vaccine strategies. This prospective study aims to evaluate immune responses and breakthrough infection in 235 infection-naïve healthcare workers up to 13–15 months after initial vaccination in two vaccine groups (108 BNT/BNT/BNT and 127 ChAd/ChAd/BNT). Immune responses were assessed using the interferon-gamma enzyme-linked immunospot (ELISPOT) assay, total immunoglobulin, and neutralizing activity through surrogate virus neutralization test at nine different time points. Both groups exhibited peak responses one to two months after the second or third dose, followed by gradual declines over six months. Notably, the ChAd group exhibited a gradual increase in ELISPOT results, but their antibody levels declined more rapidly after reaching peak response compared to the BNT group. Six months after the third dose, both groups had substantial cellular responses, with superior humoral responses in the BNT group (*p* < 0.05). As many as 55 breakthrough infection participants displayed higher neutralization activities against Omicron variants, but similar cellular responses compared to 127 infection-naïve individuals, suggesting cross-immunity. Distinct neutralization classifications (<30%, >80% inhibition) correlated with different ELISPOT results. Our study reveals diverse immune response patterns based on vaccine strategies and breakthrough infections, emphasizing the importance of understanding these dynamics for optimized vaccination decisions.

## 1. Introduction

The coronavirus disease 2019 (COVID-19), caused by the severe acute respiratory syndrome coronavirus-2 (SARS-CoV-2), remains a persistent global pandemic, posing an ongoing viral challenge, primarily due to the appearance of notable variants of concern (VOCs), including Omicron and its sub-lineage [1,2,3]. Vaccination is acknowledged as the most efficient approach to attaining herd immunity across the worldwide population. COVID-19 vaccines, including mRNA vaccines, adenoviral vector-based vaccines, inactivated virus vaccines, and subunit vaccines, have been demonstrated to significantly reduce the mortality associated with SARS-CoV-2 infection [4,5].

The protective effect of vaccination persists for 6-8 months following the second dose [6], necessitating a recommended booster 6 months after completing the primary series of vaccination [7]. The continuous emergence of SARS-CoV-2 variants has raised concerns regarding the diminishing efficacy of vaccination, thus also increasing the need for booster vaccine [8]. Some studies have demonstrated that an effective defense against SARS-CoV-2 infection or disease may not rely solely on humoral responses; rather, SARS-CoV-2-specific cellular immune responses could offer superior protection [9]. 

Cellular immunity, particularly T cell-mediated immunity, plays an important role in recognizing and controlling viral pathogens. For COVID-19, T cell-mediated immunity has shown to be associated with effective viral clearance, reduced disease severity, and protection against reinfection [10,11,12]. Compared to humoral immunity, cellular immunity has exhibited greater long-term stability after vaccination or infection [13,14]. Moreover, even though immune protection after SARS-CoV-2 infection or vaccination may wane with the emergence of novel VOC, a substantial number of T cell epitopes are conserved among VOCs, facilitating disease control beyond humoral response [14,15]. 

Nevertheless, in-depth longitudinal studies that comprehensively assess the immune response induced by currently approved vaccines in naive healthy individuals remain scarce. The contribution of vaccine-induced immunity to SARS-CoV-2 infection involving emerging variants also remains poorly defined. Moreover, research incorporating labor-intensive techniques such as enzyme-linked immunospot (ELISPOT) assays remains limited. The ELISPOT assay, based on the detection of cytokine secretion from a single cell, has proven valuable in assessing T cell response to SARS-CoV-2 [11,16].

Here, we conducted a prospective longitudinal cohort study in infection-naïve healthcare workers (HCWs) to assess the cellular response using ELISPOT assay, as well as the humoral response using spike protein-specific antibodies and surrogate virus neutralization tests (sVNT). To examine longitudinal changes, we conducted serial assessments spanning 13–15 months. Among the approved vaccines, South Korea has primarily administered a two-dose primary series involving the mRNA vaccine named BNT162b2 (Pfizer-BioNTech, referred to as BNT) and an adenoviral vector-based vaccine known as ChAdOx1 nCoV-10 (AstraZeneca, referred to as ChAd). With the emergence and rapid spread of Omicron variants, the initial strategy has been to administer BNT boosters, primarily due to vaccine-induced adverse events associated with ChAd and vaccine supply constraints [17]. Consequently, we measured immune responses at nine different times points, including before and after administration of the first and second vaccine doses (BNT/BNT or ChAd/ChAd) and after the subsequent BNT booster dose. We also analyzed immune responses to Omicron variants in samples during the Omicron epidemic via ELISPOT and sVNT assays targeting Omicron variant-specific antigens. Given the ongoing emergence of new variants and the diminishing efficacy of humoral responses, a comprehensive understanding of cellular responses to SARS-CoV-2 is necessary in order to improve SARS-CoV-2 vaccine strategies.

The primary objective of our study was to undertake a detailed characterization of the immune response and compare the two primary vaccination strategies through longitudinal analysis of cellular and humoral immunity. Our secondary aim was to compare immune responses and cross-immunity to Omicron variants between infection-naïve participants and those who experienced breakthrough infection after full vaccination.

## 2. Materials and Methods

### 2.1. Study Design and Sampling

This longitudinal prospective cohort study included 235 HCWs who received 2 doses of vaccination (2 doses of BNT or 2 doses of ChAd) followed by a BNT booster dose at Seoul St. Mary’s hospital between March 2021 and January 2022. The inclusion criteria comprised adult HCWs aged 20 years or older who had neither current nor prior diagnoses of COVID-19 based on PCR testing prior to vaccination. They received a third dose of BNT vaccination six months following the second dose and were monitored until six months after the third dose. They received a third BNT vaccination dose six months after the second dose and were under observation for another six months after the third dose. Blood samples were obtained at nine specific time points: prior to the first vaccination (T1), before the second vaccination (T2), one month after the second vaccination (T3), two months after the second vaccination (T4), three months after the second vaccination (T5), six months following the second vaccination (T6), one month after the third dose (T7), three months after the third dose (T8), and six months after the third dose (T9). The timing of the sampling, the between samplings, and the number of samples are elucidated in Figure 1. 

### 2.2. Study Groups

A total of 235 HCWs were categorized into the BNT group (*n* = 108) and ChAd group (*n* = 127) based on their homogenous primary vaccine series. Baseline characteristics are presented in Table 1. A subgroup analysis was performed on participants who encountered breakthrough infections during the study. Breakthrough infections were defined as either confirmed COVID-19 positivity via reverse transcription polymerase chain reaction (RT-PCR) or the presence of specific antibodies against the SARS-CoV-2 nucleocapsid antigen. In this prospective cohort study, a total of 182 HCWs (78 in the BNT group and 104 in the ChAd group) participated in this study up to 6 months after their third dose. SARS-CoV-2 breakthrough infections were confirmed in 14.1% (11/78) of the BNT group and 42.3% (44/104) of ChAd group. Among the participants who experienced breakthrough infections, no severe cases were reported, which includes instances of emergency room visits, hospitalizations, or fatalities. All participants provided written informed consent for the utilization of their samples in research. This study received approval from the Institutional Review Board at Seoul St. Mary’s Hospital (KC21TISI0114, KC22SISI0315).

### 2.3. SARS-CoV-2 S (Wild Type, Omicron BA.1, BA.2, BA.5) IFN-γ ELISPOT Assay

Isolation of mononuclear cells, with a specific focus on peripheral blood mononuclear cells (PBMCs), was performed by freshly extracting them from heparinized whole blood using density gradient centrifugation with a Leucosep device (Greiner bio-one International, Kremsmünster, Austria). PBMCs were cryopreserved with FBS in 10% DMSO (Sigma-Aldrich, St. Louis, MO, USA) and stored in liquid nitrogen until the ELISPOT assay was conducted. Before the ELISPOT assay, cryopreserved PBMCs were thawed, washed, and counted, and the resulting cell suspension was adjusted to a final concentration of 2.5 × 10^6^ cells/mL.

The SARS-CoV-2 ELISPOT assay utilized the BD Human IFN-γ ELISPOT kit, which was sourced from BD Biosciences in San Jose, CA, USA. The SARS-CoV-2 wild-type (WT) peptide pools comprised 15-mer sequences with an overlapping of 11 amino acids (aa), encompassing the surface spike protein. These peptide pools were obtained from Miltenyi Biotec in Bergisch Gladbach, Germany. The PepTivator^®^ SARS-CoV-2 Prot_S encompasses the immunodominant regions of the surface glycoprotein, covering the amino acid sequence domains from aa 304 to 338, 421 to 475, 492 to 519, 683 to 707, 741 to 770, and 785 to 802, and extending up to aa 885, which is the sequence end (1273). The PepTivator^®^ SARS-CoV-2 Prot_S1, on the other hand, specifically includes the N-terminal S1 domain spanning aa sequence 1–692. The PepTivator^®^ SARS-CoV-2 Prot_S+ consists of aa 689–895 of the surface glycoproteins of SARS-CoV-2. To assess Omicron variant-specific cellular immunity, we conducted additional SARS-CoV-2 ELISPOT assays using SARS-CoV-2 Prot S B.1.1.529/BA.1, BA.2, or BA.5 peptides (Miltenyi, PepTivator SARS-CoV-2 Prot_S B.1.1.529/BA.1 Mutation Pool and PepTivator SARS-CoV-2 Prot_S B.1.1.529/BA.2 Mutation Pool) (Figure 2). SARS-CoV-2 Prot_S B.1.1.529 mutation pool selectively covers the mutated regions (Appendix A). The 96-well ELISPOT plates were coated with purified NA/LE anti-human IFN-γ antibody and incubated at 4 °C overnight. After washing the plates with a coating buffer, they were incubated with 1× RPMI-1640 in 5% human serum as a blocking solution for 30 min. PBMCs (2.5 × 105 cells/well) were then stimulated with peptide (2 μg/mL) overnight in a 36 °C 5% CO_2_ incubator. PBMCs were subjected to additional stimuli, including phorbol 12-myristate 13-acetate (PMA)/ionomycin as a positive control and RPMI medium as a negative control. After washing the microplate with a solution of distilled water and PBS pH 7.2 containing 0.05% Tween 20 from Genetech in Hanam, Republic of Korea, we introduced biotinylated anti-human IFN-γ detection antibody and allowed it to incubate at room temperature for a duration of 2 h. Following this, we executed the subsequent steps as per the manufacturer’s guidelines, including the utilization of streptavidin-HRP and the AEC substrate. The quantification of spots on the dried microplate was accomplished using the AID iSpot ELISpot flourospot reader from AID in Strassberg, Germany. Spot numbers were determined by subtracting the count of spots in the negative control from the average count of spots in the test wells. The results were expressed as spot forming units (SFU) per 106 PBMCs, with a test cut-off of 10 SFU/106 PBMCs established for each antigen. 

### 2.4. SARS-CoV-2 Binding Antibody Assay 

We assessed anti-SARS-CoV-2 antibodies using the Roche Elecsys Anti-SARS-CoV-2 S chemiluminescent immunoassay, conducted on a Roche Cobas e-801 instrument (Roche Diagnostics, Basel, Switzerland). This test is designed for the quantitative measurement of total antibodies against the receptor-binding domain (RBD) of the SARS-CoV-2 S antigen. A threshold of 0.8 U/mL was used, and the measurable range extended from 0.4 to 250 U/mL according to the manufacturer’s guidelines. Samples exceeding 250 U/mL were subject to retesting after dilution. The results were reported in binding antibody units per mL (BAU/mL), which were standardized to WHO international standards for anti-SARS-CoV-2 immunoglobulin, using a conversion factor of 1.028. To identify cases of breakthrough SARS-CoV-2 infection, we also employed a Roche assay designed to detect anti-SARS-CoV-2 nucleocapsid antibodies. A cutoff value of 1.0 U/mL was utilized for the Elecsys anti-SARS-CoV-2-N assay.

### 2.5. SARS-CoV-2 Surrogate Virus Neutralization Test

We assessed the percentage of neutralizing antibodies’ inhibition using the SARS-CoV-2 surrogate virus neutralization test (sVNT) from GenScript, located in Piscataway, NJ, USA. The sVNT is an enzyme-linked immunosorbent assay (ELISA) based on competitive principles. Its purpose is to measure the prevention of interactions between the wild-type angiotensin-converting enzyme 2 (ACE2) receptor and the receptor-binding domain (RBD) of the SARS-CoV-2 virus. Following the manufacturer’s guidelines, results were interpreted using a threshold of ≥30% inhibition. Additionally, for samples collected 4–6 months after booster vaccination, including those from participants who had experienced breakthrough infections, we further evaluated neutralization efficacy using the sVNT assay, specifically targeting the SARS-CoV-2 Omicron variant (GenScript). 

### 2.6. Statistical Analysis

We reported continuous data as the median and interquartile range. Nonparametric quantitative results were subjected to statistical analysis using the Mann–Whitney U test and Kruskal–Wallis test. Categorical data were presented as counts and percentages and were subjected to analysis using either the Chi-squared test or Fisher’s exact test. Correlations between quantitative values obtained from different assays were determined through Spearman’s rank correlation analysis. We conducted statistical analyses using Prism version 10.0.1 for Windows (GraphPad, San Diego, CA, USA) or MedCalc statistical software version 22.007 (MedCalc Software Ltd., Ostend, Belgium). Statistical significance was denoted as follows: * *p* < 0.05, ** *p* < 0.01, *** *p* < 0.001, **** *p* < 0.0001. 

## 3. Results

### 3.1. SARS-CoV-2-Spike-Specific ELISPOT Response after Vaccination in Infection-Naïve Participants

Longitudinal changes in cellular immune response by IFN-γ ELISPOT assays targeting SARS-CoV-2 S, S1, and S+ peptide pools are shown in Figure 3A. One month after the second dose, the BNT group exhibited an increase in T cell response (S: 12.0-fold; S1: 13.2-fold; S+: 8.0-fold) compared to the levels observed one month after the first dose (T2). However, there was a significant decrease over time until 6 months after the second dose (*p* < 0.01). In the ChAd group, T cell responses at T3 gradually increased (S: 1.1-fold; S1: 1.4-fold; S+: 1.5-fold) compared to those observed at one month after the first dose (T2), and they peaked significantly at 2 months. However, substantial T cell responses were observed for up to 6 months in both groups, and ELISPOT results exhibited no significant difference between the BNT and ChAd cohorts at 6 months after the second dose (*p* > 0.05). 

Similarly, one month after the third BNT dose (T7), the BNT group showed a significant increase in T cell response (S: 3.9-fold; S1: 3.4-fold; S+: 3.4-fold) compared to the period before the third vaccination at 6 months after the second dose (T6). The ChAd group also demonstrated an increase in T cell response (S: 2.3-fold; S1: 2.2-fold; S+: 1.5-fold) compared to the previous sampling period (T6). However, at one month after the third dose (T7), the ChAd group showed a significantly lower response compared to the BNT group (*p* < 0.05). However, the induced cellular response targeting S, S1, and S+ antigens significantly decreased over time in both cohorts. Cellular responses at six months after the third dose were similar to those at six months after the second dose (BNT group; 0.9–1.3-fold, ChAd group; 1.0–1.6-fold). Ultimately, ELISPOT results revealed no significant difference between the BNT and ChAd cohorts at six months after the third dose (*p* > 0.05).

### 3.2. Positivity Rate of SARS-CoV-2-Spike-Specific ELISPOT after Vaccination in Infection-Naïve Participants 

We defined >10 SFU/10^6^ PBMCs as a positive ELISPOT response, and changes in positivity rates are shown in Figure 3B. For 1–6 months after the second dose, the BNT group elicited a positive response of ELISPOT to S, S1, and S+ in 66.7–85.4%, 71.3–86.4%, and 65.6–81.6% of the samples, respectively, and showed the highest positive rate at 2 months after the second dose. The ChAd group showed a positive response of 79.5–89.1%, 79.5–88.2%, and 70.1–83.2% for the S, S1, and S+ ELISPOT assays after the second dose, respectively. Six months after second dose, the positive rate of S and S1 ELISPOT results was higher in the ChAd group than in the BNT group (*p* < 0.05).

The positive rate of ELSIPOT results at 1 month after the third dose was higher in the BNT group than in the ChAd group (BNT group: 94.7–100%; ChAd group: 71.7–78.3%, *p* < 0.05). However, from 3 to 6 months after the third dose, both groups showed a positive response rate of more than 90% for ELISPOT to S and S1, and no significant difference was found between the two cohorts (*p* > 0.05).

### 3.3. Anti-SARS-CoV-2-S Binding Antibody and Neutralizing Antibody Responses after Vaccination in Infection-Naïve Participants

The changes in humoral immune responses were analyzed from before vaccination to after the third vaccination doses (Figure 4A). One month after the second dose, the levels of SARS-CoV-2-specific binding antibodies increased in both groups (median (IQR) (BAU/mL); BNT: 1495.2 (985.9–1901.8); ChAd: 780.3 (508.4–1280.1)). Neutralizing antibody levels were measured to be 96.0 (95.0–97.0)% in the BNT group and 85.0 (63.8–94.0)% in the ChAd group. Both binding antibody and neutralizing antibody levels peaked at 1–2 months after the second dose and significantly waned over 6 months in both groups (*p* < 0.0001). The BNT group showed higher S-specific binding antibody levels and neutralizing antibody levels compared to those in the ChAd group at 1 month and 6 months after second doses (*p* < 0.0001) (Figure 4). 

After the third BNT dose booster, binding antibody levels increased significantly to 14,042.5 (10,002.4–18,791.8) BAU/mL in the BNT group and 10,506.2 (7072.6–15,358.3) BAU/mL in the ChAd group, and neutralizing antibody levels also increased to 98.0% inhibition in both groups. Like the kinetics of humoral responses after the second dose, both binding antibody and neutralizing antibody levels decreased significantly over 6 months after third dose (*p* < 0.0001). The ChAd group revealed a faster decrease in both binding and neutralizing antibody levels. However, the binding antibody levels at 6 months after the third dose were 9.2-fold and 7.1-fold higher in the BNT and ChAd groups, respectively, compared to 6 months after the second dose. Six months after the third dose, the median % inhibition of neutralizing antibody levels was 97% in the BNT group and 94% in the ChAd group. At all the following points, the BNT group showed increased humoral response compared to the ChAd group, which was statistically significant at 6 months after third dose (*p* < 0.0001). 

Changes in positivity rates of humoral responses are shown in Figure 4B. After vaccination, SARS-CoV-2-specific binding antibody results were positive using the manufacturer’s provided cut-off (0.8 U/mL) at all time points except for three participants (two in the BNT and one in the ChAd group) at 1 month after the second dose. sVNT showed higher positive results in the BNT group at 1, 2, 3, and 6 months after the second vaccination compared to the ChAd group (89.9–100% vs. 59.2–97.4%). However, after the third dose, only one participant in the ChAd group had negative neutralizing antibody (19% inhibition), and there was no significant difference in the neutralizing antibody positive rate between the two groups (*p* > 0.05).

### 3.4. Cellular and Humoral Immune Responses against WT and Omicron Variants in Participants with Breakthrough Infection

Using the specimens collected 6 months after thr third dose, cellular and humoral immune responses were compared according to SARS-CoV-2 breakthrough infection status (Figure 5). Breakthrough infections occurred between January 2022 and May 2022, aligning with the prevalence of Omicron strains in the study area. To assess cellular responses to SARS-CoV-2 WT and Omicron variants, an ELISPOT assay using SARS-CoV-2 WT, BA.1, BA.2, and BA.5 peptide pools was performed. Participants who experienced breakthrough infection showed higher SARS-CoV-2 WT-specific-ELISPOT results compared to the infection-naïve participants (*p* < 0.0001). However, there was no statistically significant difference in ELISPOT results for SARS-CoV-2 variants (BA.1, BA.2, and BA.5) between infection-naïve and breakthrough infections (Figure 5A). Furthermore, no statistically significant differences in ELISPOT results were observed between the two vaccine groups in both infection-naïve individuals and those with breakthrough infections (*p* > 0.05). Most participants, whether infection-naïve or with breakthrough infection, had positive cellular responses to stimulation of WT and Omicron peptide pools. 

We also measured SARS-CoV-2 neutralizing antibody levels against SARS-CoV-2 WT and Omicron variants (SARS-CoV-2 BA.1, BA.2, and BA.4–5) in infection-naïve and breakthrough infection participants using available samples at 6 months after the third dose. In contrast to the cellular immune response, infection-naïve participants showed significantly lower neutralization activity against both WT and Omicron variants compared to participants with breakthrough infection. In addition, the level of neutralizing activity against Omicron variants varied depending on the strain. Among infection-naïve participants, 75.8% and 20.8% showed negative neutralizing activity (<30% inhibition) against BA.1 and BA.4–5 Omicron variants, respectively. When comparing the two vaccine groups, the BNT group demonstrated a tendency to display higher neutralizing activity against Omicron variants than the ChAd group, although there was no statistically significant difference (*p* > 0.05) (Figure 6).

Next, we assessed whether the immune response to WT antigens at each sampling time point could predict Omicron breakthrough infection 3–6 months after the third dose. Changes in immune responses were compared between infection-naïve and breakthrough-infected participants up to 6 months after the third dose. In both groups, neither cellular nor humoral responses to WT at each sampling time point were predictive of breakthrough infection (Appendix A).

### 3.5. Correlation between Cellular and Humoral Immune Response according to the Assay Methods

When analyzing the correlation between cellular immune responses measured by ELISPOT for S, S1, or S+ antigens, we found a strong correlation between the results (r = 0.81–0.89) in both groups. Similarly, the humoral responses measured via total binding antibody assays were strongly correlated with neutralization activities (r = 0.89–0.91), and there was no difference between the two vaccine groups. However, cellular responses via ELISPOT assays and humoral responses were not correlated in both the BNT and ChAd groups (Appendix A). 

Next, we compared the ELISPOT results according to the classified sVNT results into three grades (strong: >80%; moderate: 30–80%; weak: <30%). The samples with neutralization activity > 80% inhibition had higher ELISPOT results than other samples (Appendix A). 

## 4. Discussion

Previous studies have reported the effect of robust cellular and humoral immune responses following booster vaccination [18,19]. However, it is still not well understood how long the vaccine-induced immune response lasts and how it changes with booster vaccination. Here, in a prospective longitudinal study, the two primary series vaccination groups, including mRNA vaccine (BNT/BNT) and vector vaccine (ChAd/ChAd), were evaluated up to 6 months after the third BNT vaccination. 

The cellular response, as assessed via the ELISPOT assay, reached its highest level at 1–2 months after the second or third dose and gradually declined over the course of 6 months in both the BNT and ChAd vaccine groups. When comparing the cellular immune response one month after the booster vaccination, the BNT group exhibited a higher T-cell immune response compared to the ChAd group. Notably, the ChAd group showed a gradual increase in cellular response over time, surpassing the BNT group. 

However, it is worth mentioning that ELISPOT results weakened between 3 to 6 months after the second or third dose, and by 6 months after the third BNT dose, there were no significant differences in cellular responses between the BNT and ChAd cohorts. These findings align with previous research indicating that a heterologous vaccination approach involving ChAd-BNT results in a robust cellular immune response [12,20,21]. Additionally, these results confirm previous reports that the heterologous vaccine group and the homologous vaccine group, including mRNA vaccines, showed similar immune responses based on different analysis platforms [12]. Regarding the positivity rate of cellular responses, defined by a cutoff of >10 SFU/10^6^ PBMC, more than 90% of participants in both groups exhibited a positive response at 3 to 6 months after the third dose, and no significant differences in the positivity rate were observed between the two groups. 

Humoral immune response, measured by total binding antibody and neutralizing activity, also reached its peak level at 1–2 months following the second or third vaccination and then declined rapidly over the 6-month period post-vaccination. The decline in humoral immunity was observed in both vaccine groups, but the ChAd group exhibited a faster decrease in both binding and neutralizing antibody levels, which aligns with findings from a previous study [22]. Additionally, when comparing humoral responses between the two groups, the BNT group tended to have a higher humoral response than the ChAd group at all sampling points. This difference in humoral response was statistically significant 6 months after the third vaccination, consistent with previous research results [19]. 

Up to 6 months after the third dose, SARS-CoV-2 breakthrough infections occurred more frequently in the ChAd group compared to the BNT group (*p* < 0.001). In the present study, breakthrough infections occurred most often between 4 and 6 months after the third dose in the ChAd group and at 6 months after the third dose in the BNT group, coinciding with the dominance of Omicron strains in the study area. The longer exposure to Omicron strains in the ChAd group compared to the BNT group may be one of the reasons for the higher frequency of breakthrough infections. The current study was conducted among healthy participants, most of whom demonstrated sustainable and durable cellular immunity. The finding that none of the participants who developed breakthrough infections reported serious events aligns with prior research indicating that cellular immunity contributes to preventing the development of severe disease in individuals with breakthrough infections [23,24,25,26]. Additionally, this supports previous reports indicating that a BNT booster is likely to offer protection against serious and fatal diseases [17,27].

Following breakthrough infection, the ELISPOT response to WT spike peptides significantly increased in comparison to infection-naïve participants (*p* < 0.0001). This finding aligns with previous suggesting a robust cellular response post-breakthrough infection [25]. However, the ELISPOT results targeting the Omicron strains (BA.1, BA.2, and BA.5) remained unaffected by the experience of breakthrough infection experience, demonstrating a similar cellular response as infection-naïve participants. In our study, most infection-naïve participants displayed positive cellular responses against Omicron variants as well as the SARS-CoV-2 WT antigen at 6 months after the third BNT booster, regardless of their experiencing breakthrough infection or following different vaccine strategies. These findings support the previous report that SARS-CoV-2 infection can be as effective as vaccination [28]. Interestingly, after a breakthrough infection, presumably by the Omicron lineage within our study population, most participants exhibited higher cellular responses to WT antigens than to Omicron antigens (BA.1, BA.2, and BA.5). This finding is consistent with the concept of a back-boosting effect on immunologic imprinting. Our cohorts initially comprised infection-naïve participants, all of whom were initially exposed to the SARS-CoV-2 spike through vaccines based on the ancestral spike. Subsequent exposure to more antigenically advanced strains, such as the Omicron variant, may have led to increased antibody titers against the ancestral antigens [29]. Our results support previous reports indicating that vaccination is associated with cross-protection against viral infection and induces cross-reactive memory T cell immunity specific to viral epitopes, providing protection against SARS-CoV-2 infection [14,30]. 

In contrast to the cellular responses, when we assessed infection-naïve participants using sVNT against Omicron antigens at 6 months after the third dose, the level of neutralizing activity against Omicron variants was significantly lower compared to post-breakthrough-infected participants. This diminished Omicron neutralizing activity was more pronounced in the specific sub-lineage BA.1, especially among participants in the ChAd group. This reduced neutralization against BA.1 may be related to the findings reported by Yang Y. et al., who reported that BA.1 infection induced poor cross-neutralization against other variants compared to BA.2 [31]. Combined with our observations, it suggests that BA.1 exhibits limited neutralization against other variants. It is also worth noting that the homologous BNT group tended to have stronger neutralizing activity against Omicron subvariants in comparison to individuals in heterologous ChAd/ChAd/BNT groups. Whether homologous three-dose BNT can induce broader neutralizing activity compared to heterologous vaccination needs to be elucidated in future research.

Our findings also support previous reports indicating that the humoral response diminished rapidly following vaccination and that assessing vaccine effectiveness solely based on humoral immunity may be insufficient [23,32]. When comparing the two vaccine groups, the BNT group displayed a tendency toward higher neutralizing activity against Omicron variants and a decreased occurrence of breakthrough infection compared to the ChAd group. This finding is consistent with previous suggestions that lower humoral immune responses against specific strains after the third dose may be linked to a higher frequency of breakthrough infections [12]. Furthermore, it supports the previous report that BNT vaccines appear to be superior to ChAd vaccines in inducing neutralizing antibodies against Omicron variants after booster vaccination [33]. 

Next, we analyzed correlations of results from different assays for humoral or cellular responses. When we compared cellular responses via ELISPOT against different S- antigens, a strong correlation was observed. Similarly, the humoral responses assessed through total binding antibody assays exhibited a strong correlation with neutralizing activities in both groups, consistent with findings in previous studies [34,35,36,37]. However, we found no correlation between quantitative ELISPOT results and total binding antibody levels or sVNT results, which is in line with previous study [38]. When we categorized sVNT results into three grades (strong: >80%, moderate: 30–80%, weak: <30%), we observed a significant difference in ELISPOT results among the different sVNT grades. Because labor-intensive cellular response assessments may not be feasible for routine use, the estimation of serologic response becomes crucial for evaluating protection against infection and assessing vaccine effectiveness. Larger cohorts are necessary in order to establish threshold levels of serological response across different vaccine strategies and in diverse populations.

This study has several limitations. First, this study lacks a sample of patients with severe breakthrough infections or individuals studied beyond the 6 months after the third vaccination. Second, all participants were healthy individuals from a single center, and the BNT group had a higher proportion of male participants, potentially introducing bias into the study’s results. Future long-term studies should encompass more diverse cohorts, including individuals with varying disease profiles related to disease severity in infected patients. Additionally, neutralizing antibodies against new strains, including the Omicron variant, were not tested in all collected samples. There were also significant differences in vaccine intervals between study groups. The ChAd group had longer intervals for the first and second doses (median 77 days) compared to the BNT group (median 21 days). Similarly, third-dose intervals differed, with a median of 165 days for ChAd and 210 days for BNT. These different vaccine schedules may have contributed to the observed discrepancies in immune responses between the two groups.

Despite these limitations, our study demonstrated distinct immune response patterns among participants based on vaccine strategies and breakthrough infection status. While substantial humoral and cellular immunity was established following the third BNT booster after a two-dose primary series with either BNT or ChAd, our findings highlight importance of recognizing the diversity in immune responses among participants and the clear distinction between humoral and cellular responses. 

This prospective longitudinal study offers comprehensive insights into the cellular and humoral responses following the administration of the third BNT booster in BNT-primed and ChAd-primed individuals. It emphasizes that not only humoral but also cellular immune parameters are essential for comprehending the adaptive immune response against SARS-CoV-2. We anticipate that this comprehensive investigation will provide valuable evidence to inform future vaccine-related decisions.

## Figures and Tables

**Figure 1 vaccines-11-01613-f001:**
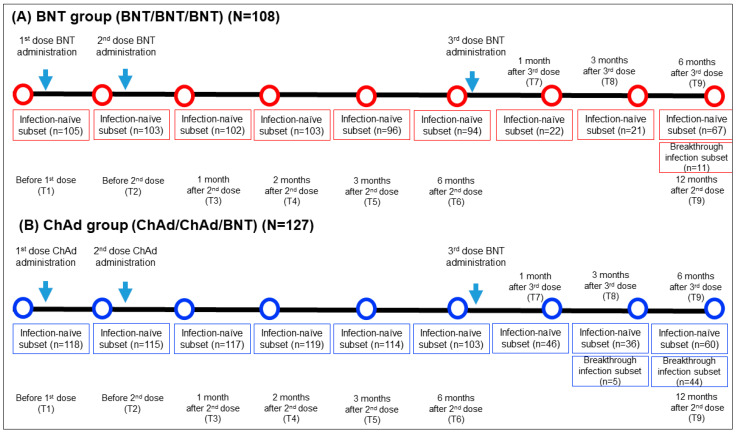
Comprehensive sampling schedule and the participant count for both the BNT (**A**) and ChAd (**B**) groups. N, total number of participants in each group; n, number of participants who had blood sample collected at specific timepoint.

**Figure 2 vaccines-11-01613-f002:**
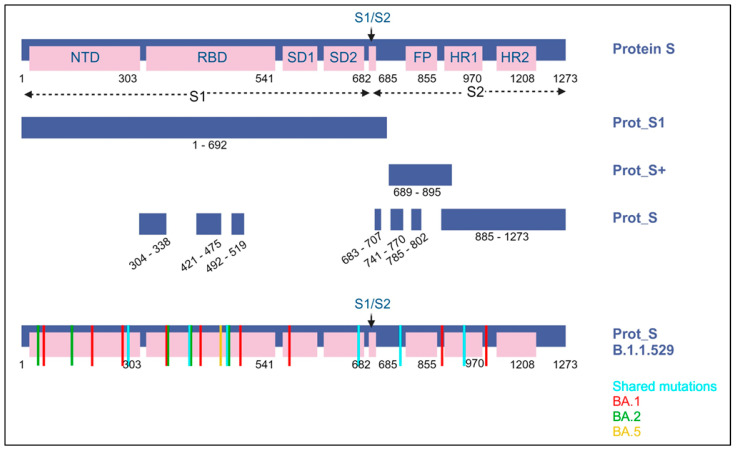
A visual representation of the SARS-CoV-2 spike protein and the protein sections targeted by the peptide pools (including protein S1, S+, and S) employed in ELISPOT assays. Within the B.1.1.529 lineage (Omicron variant), it is identified as Omicron BA.1, BA.2, and BA.5. Blue indicates shared substitutions; red, Omicron BA.1; green, BA.2; and yellow, BA.5.

**Figure 3 vaccines-11-01613-f003:**
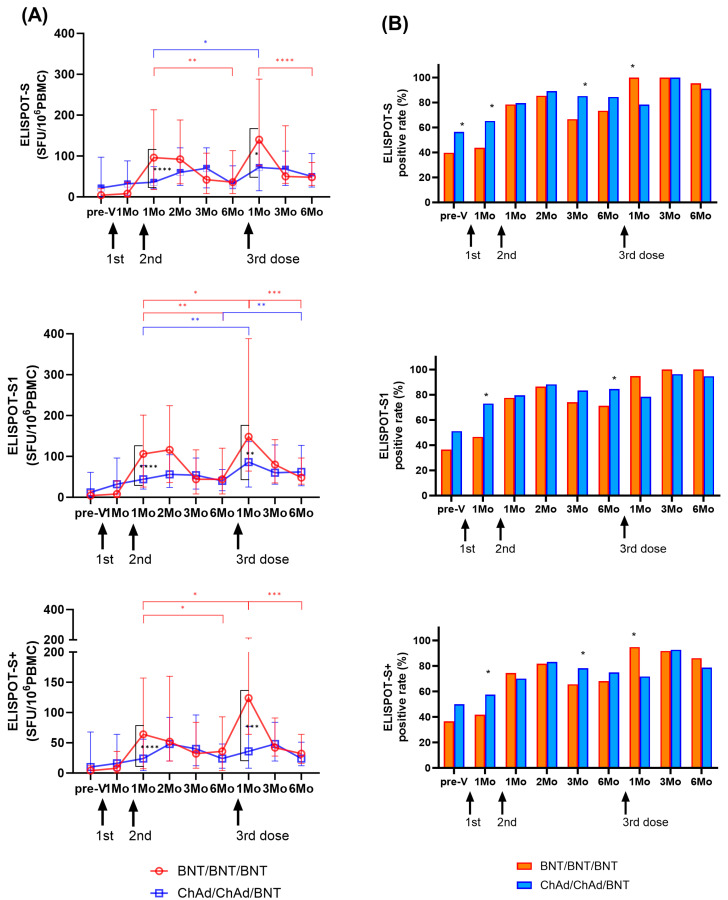
Longitudinal changes in cellular immune responses in infection-naïve participants. SARS-CoV-2-specific T cell responses against S, S1, and S+ antigens were assessed using ELISPOT. Results are presented in spot forming unit (SFU)/10^6^ peripheral blood mononuclear cells (PBMCs) (**A**) and as a positive rate (**B**). The statistical significance was determined through the Mann–Whitney test (**A**), with significant change during the study period indicated by ticked lines (red: BNT group; blue: ChAd group). A comparison of the positive rates between the two groups was conducted using the Chi-square test or Fisher’s exact test (**B**). * *p* < 0.05; ** *p* < 0.01; *** *p* < 0.001; **** *p* < 0.0001.

**Figure 4 vaccines-11-01613-f004:**
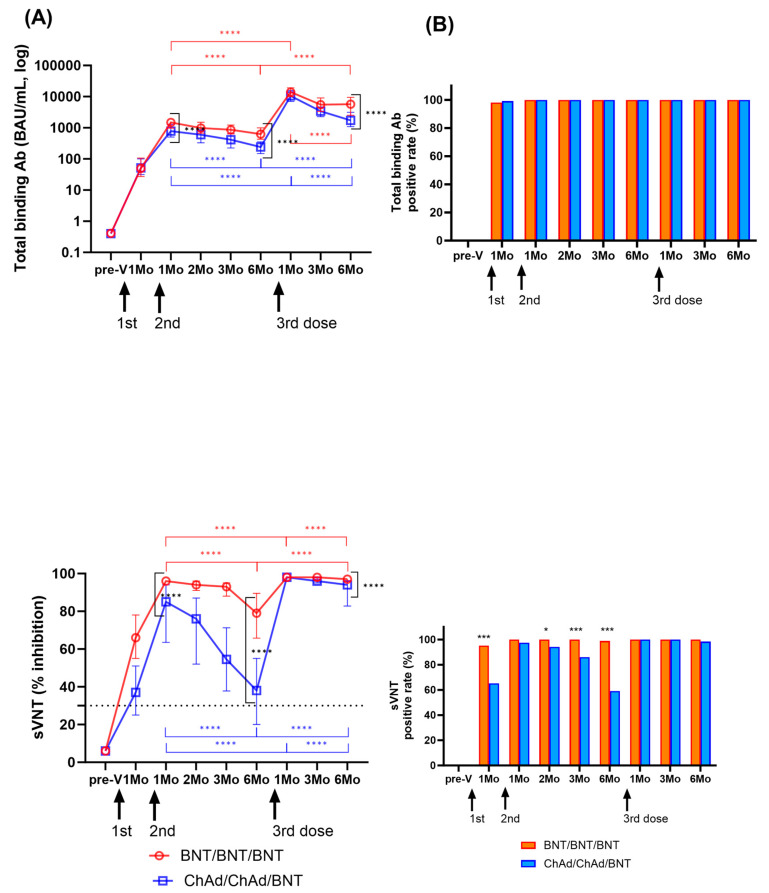
Longitudinal changes in humoral immune response were assessed by measuring binding antibody levels and neutralization activity among participants who had not been previously infected with SARS-CoV-2. Panel (**A**) displays SARS-CoV-2 spike-specific total binding antibody levels and the percentage of inhibition as median values with the interquartile range. Statistical significance was evaluated using the Mann–Whitney test, and significant changes observed during the study period are indicated by marked lines (red: BNT group; blue: ChAd group). In Panel (**B**), the positivity rates for total binding antibodies (>0.8 IU/mL) and neutralization activity (≥30%) at each sampling time are also presented. A comparison of the positivity rates between the BNT and ChAd groups was conducted using the Chi-square test or Fisher’s exact test. The assay cut-off is depicted as a horizontal dashed line. Significance levels are denoted as follows: * *p* < 0.05; *** *p* < 0.001; **** *p* < 0.0001.

**Figure 5 vaccines-11-01613-f005:**
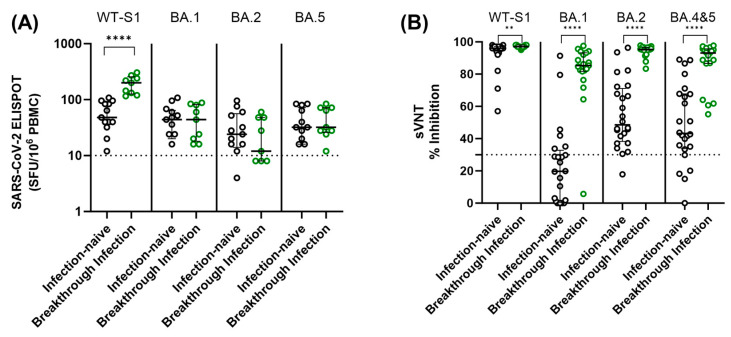
Comparison of SARS-CoV-2 ELISPOT results (**A**) and neutralization activity (**B**) against both WT and omicron variants among the individuals without prior infection and those with breakthrough infections. Samples were obtained from participants six months after the third dose. The assay cut-off is indicated by a horizontal dashed line. Statistical significance is denoted as follows: ** *p* < 0.01; **** *p* < 0.0001 via Mann–Whitney U test.

**Figure 6 vaccines-11-01613-f006:**
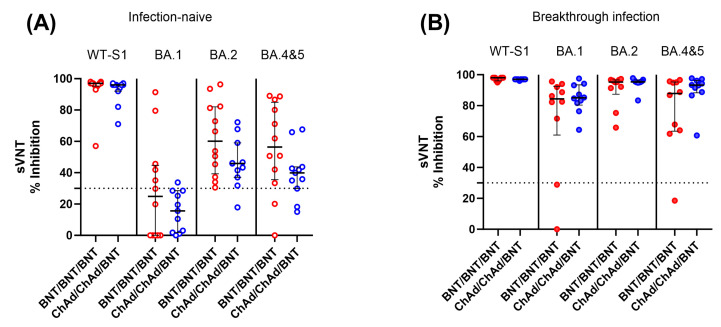
Comparison of SARS-CoV-2 neutralization activity between the BNT/BNT/BNT group and the ChAd/ChAd/BNT group against both the WT and omicron variants in infection-naïve individuals (**A**) and participants with breakthrough infection (**B**). Samples were collected from participants 6 months after the third dose. The assay cut-off is represented as a horizontal dotted line. There was no statistically significant difference between the two vaccine groups (*p* > 0.05).

**Table 1 vaccines-11-01613-t001:** Demographic characteristics of the BNT and ChAd groups.

	BNT Group (*n* = 108)	ChAd Group (*n* = 127)	*p*-Value
Age, years			
Mean ± SD	40 ± 11	40 ± 10	NS
Median (range)	37 (25–72)	37 (24–59)	NS
Sex, *n* (%)			*p* = 0.008
Male	45 (41.7)	32 (25.2)	
Female	63 (58.3)	95 (74.8)	
Vaccination schedule			
First to second dose interval (days)			
Median (range)	21 (21–22)	77 (70–99)	
Second to third interval (days)			
Median (range)	210 (200–297)	165 (154–206)	
Breakthrough infection, % *	14.1 (11/78)	42.3 (44/104)	*p* < 0.0001
Infection with positive confirmatory tests			
	10	38	
Infection with positive SARS-CoV-2 nucleocapsid antibody assay			
	1	6	
Time of breakthrough infection, *n*			
Within 1 month following the third dose administration			
	0	0	
Between 1–3 months’ window following the third dose administration			
	0	5	
Between 3–6 months’ window following the third dose administration			
	11	39	

* Percentages calculated for participants eligible for blood sample collection 6 months after third dose.

## Data Availability

The data presented in this paper are available on request from the corresponding author.

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
