# Peer review of "Longitudinal Analysis of SARS-CoV-2-Specific Cellular and Humoral Immune Responses and Breakthrough Infection following BNT162b2/BNT162b2/BNT162b2 and ChAdOx1/ChAdOx1/BNT162b2 Vaccination: A Prospective Cohort in Naive Healthcare Workers"

_vaccines, 2023, doi:10.3390/vaccines11101613_

Round 1

Reviewer 1 Report

Minor issues:

1. Line 40 should mention other types of COVID19 vaccines, such as inactivated vaccines and subunit vaccines.

2. Line 147 should clearly define the peptide pools used for Omicron, are they Pro-S pool used for Wuhan-Hu-1 aa 304-338,421-475, 492-519, 683-707, 741-770, 785-802 and 885-1273 plus mutated peptides for Omicron? Better have a table to list all Omicron specific peptide sequences.

3. Line 213-228, the fold changes are versus what? baseline, before 2nd vaccination, 3rd vaccination, etc.?

4. Line 216, 6 month ELISPOT did not show "robust" response since there are only ~50 spots/million cells on average. Should also change the "robust" wording in Abstract.

5. Line 308, what was the strain for the breakthrough infection? Omicron? Should mention either lab test confirmed Omicron or infection occurred during Omicron wave, e.g. beginning of 2022 etc.

Author Response

Please find the uploaded file (Answer to the Reviewer 1)
We really appreciate all your comments and we took them into account in our revision.

Reviewer 2 Report

The choice of topic is timely and raises important questions about vaccination strategies. The methods used have been chosen appropriately. The interpretation of the results is clear and well presented. The discussion of the results is adequate and provides a good basis for the design of future vaccination strategies.

Author Response

Thank you for your kind words and feedback. We sincerely appreciate your positive assessment of the value of our manuscript for future vaccination strategies.

Reviewer 3 Report

Review of the Article:

Longitudinal analysis of SARS-CoV-2 specific cellular, humoral responses and breakthrough infection following BNT162b2/BNT162b2/BNT162b2 and ChAdOx1/ChAdOx1/BNT162b2 vaccination: a prospective cohort in naive healthcare workers.

This prospective study assessed the immune responses and breakthrough infections in 235 healthcare workers who received COVID-19 vaccinations. The participants were divided into two vaccine groups: BNT/BNT/BNT and ChAd/ChAd/BNT. The study measured immune responses using various assays at multiple time points.

The authors found that 1) Both vaccine groups showed peak immune responses one to two months after the second or third dose, followed by gradual declines over six months. 2) The ChAd group had a gradual increase in immune responses over time, but their antibody levels decreased more quickly after reaching the peak compared to the BNT group. 3) Six months after the third dose, both groups had strong cellular responses, with the BNT group having better humoral (antibody-related) responses. 4) Among participants with breakthrough infections, their neutralization activities against Omicron variants were higher than in infection-naïve individuals, suggesting some level of cross-immunity. 5) Different levels of neutralization activity were associated with varying ELISPOT results.

Overall, this study highlights diverse patterns of immune responses based on vaccine strategies and breakthrough infections. The authors consider that understanding these dynamics is crucial for making informed decisions about vaccination strategies.

Minor observations:

An exhaustive review of the language used is necessary, it needs to be proofread for better understanding.

Methods:

In 2.2 Study groups, there are two redundant paragraphs consider rephrase this item.

Breakthrough infection was defined based on either confirmed COVID-19 positivity or the presence of SARS-CoV-2 nucleocapsid antigen-specific antibodies. At each sampling time point, we determined SARS-CoV-2 infection status by considering clinical history and the SARS-CoV-2 nucleocapsid  antibodies in the collected samples. Furthermore, during each visit, participants provided information regarding their prior vaccination history and any previous instances of COVID-19 infection. In this prospective cohort study, a total of 182 HCWs (78 BNT group 116 and 104 ChAd group) participated in the study up to 6 months after the 3rd dose. SARS- CoV-2 breakthrough infection were confirmed either by reverse transcription polymerase chain reaction (RT-PCR) testing or by antibody binding to SARS-CoV-2 nucleocapsid in 14.1% (11/78) of the BNT group and 42.3% (44/104) of ChAd group.”

Results:

The following paragraph is a little confusing. Consider rephrasing it:

Similarly, one month after the 3rd BNT dose, the BNT group showed a significant increase in T cell response (S: 3.9-fold, S1: 3.4-fold, S+: 3.4-fold). The ChAd group displayed an increase in T cell response (S: 2.3-fold, S1: 2.2-fold, S+: 1.5-fold) 1 month after the 3rd BNT dose but showed significantly lower response compared the BNT group (P < 0.05).

Items 3.4 and 3.5

The authors referred to Figures 1S, 2S and 3S but it seems they are referring to Figures S1, S2 and S3, please correct the typos.

Supplementary material:

Figure S2 legend. Consider rephrasing since a paragraph is repeated almost entirely.

“Figure S2. Correlation between SARS-CoV-2 Spike specific immune responses. Correlation between SARS-CoV-2-Spike specific cellular response measured by ELISPOT and humoral responses assessed by total immunoglobulin and neutralizing antibody assay (sVNT) in the BNT (A) and the ChAd (B) groups.”

Major Observations:

There are three main aspects that the article is not tackling that are essential for the analysis they are trying to do:

1) One of the most interesting analysis that this article is not showing, is the comparison amongst WT and omicron variants separating the groups not only by infection-naïve and breakthrough infection but also it is necessary to include separating the groups by immunization protocols (BNT or ChAd).

A new figure 5 including this in the context of the analysis will allow the authors to validate some statements that otherwise seem a bit inferential. For instance, the authors mentioned in the Discussion:

“In the present study, breakthrough infections occurred most often between 4 and 6 months after the third dose in the ChAd group and at 6 months after the third dose in the BNT group, coinciding with the dominance of Omicron strains in the study area. The higher frequency of breakthrough infections in the ChAd group may be due to their longer exposure to Omicron strains compared to the BNT group.”

2) Another aspect that is not discussed, is the intervals between doses that are quite different between both groups. Specifically, the first to second dose median intervals are 21 in BNT and 77 in ChAd while the second to third dose median intervals are 210 in BNT and 165 in ChAd. Could these differences be influencing the variability observed in both groups?

It’s necessary for the authors to address this item.

3) Finally a crucial aspect that is not discussed enough, is the fact that the third dose of the ChAd group is the BNT vaccine. This could be generating a different immune response, meaning that this vaccine is not acting as a booster. This aspect also needs to be addressed thoroughly in the article.

An exhaustive review of the language used is necessary, it needs to be proofread for better understanding.

Author Response

Please find the uploaded file (Answer to the Reviewer 3)
We really appreciate all your comments and we took them into account in our revision
